# High Serum Phosphate Is Associated with Cardiovascular Mortality and Subclinical Coronary Atherosclerosis: Systematic Review and Meta-Analysis

**DOI:** 10.3390/nu16111599

**Published:** 2024-05-24

**Authors:** Carolina Torrijo-Belanche, Belén Moreno-Franco, Ainara Muñoz-Cabrejas, Naiara Calvo-Galiano, José Antonio Casasnovas, Carmen Sayón-Orea, Pilar Guallar-Castillón

**Affiliations:** 1Department of Preventive Medicine and Public Health, Universidad de Zaragoza, 50009 Zaragoza, Spain; carolinatorrijob@gmail.com (C.T.-B.); ainaramunozc@gmail.com (A.M.-C.); 2Instituto de Investigación Sanitaria Aragón, Hospital Universitario Miguel Servet, 50009 Zaragoza, Spain; ncalvo@unizar.es (N.C.-G.); jacasas@unizar.es (J.A.C.); 3CIBERCV (CIBER de Enfermedades Cardiovasculares), 28029 Madrid, Spain; 4Department of Medicine, Psychiatry and Dermatology, Universidad de Zaragoza, 50009 Zaragoza, Spain; 5Department of Preventive Medicine and Public Health, University of Navarra-IDISNA, 31008 Pamplona, Spain; msayon@unav.es; 6CIBERobn (CIBER Fisiopatología de la Obesidad y Nutrición), 28029 Madrid, Spain; 7Department of Preventive Medicine and Public Health, School of Medicine, Universidad Autónoma de Madrid, 28029 Madrid, Spain; mpilar.guallar@uam.es; 8CIBERESP (CIBER de Epidemiología y Salud Pública), 28029 Madrid, Spain; 9IMDEA-Food Institute, CEI UAM + CSIC, Carretera de Cantoblanco 8, 28049 Madrid, Spain

**Keywords:** serum phosphate, cardiovascular mortality, cardiovascular disease, subclinical coronary atherosclerosis, meta-analysis

## Abstract

(1) Background: Cardiovascular diseases (CVDs) are the leading cause of mortality worldwide. The aim of the study was to examine the existing published results of the association between elevated serum phosphate concentrations and cardiovascular mortality, along with the CVD incidence and subclinical coronary atherosclerosis, in primary prevention among non-selected samples of the general population. (2) Methods: A systematic review and meta-analysis were carried out using literature obtained from PubMed, SCOPUS, and the Web Of Science until March 2024 and following the PRISMA guidelines. Relevant information was extracted and presented. Random and fixed effects models were used to estimate the pooled odds ratio (OR) and hazard ratio (HR) with their 95% coefficient interval (CI), and I^2^ was used to assess heterogeneity. (3) Results: Twenty-five studies met our inclusion criteria and were included in the meta-analysis (11 cross-sectional and 14 cohort studies). For cardiovascular mortality, which included 7 cohort studies and 41,764 adults, the pooled HR was 1.44 (95% CIs 1.28, 1.61; I^2^ 0%) when the highest versus the reference level of serum phosphate concentrations were compared. For CVDs, which included 8 cohort studies and 61,723 adults, the pooled HR was 1.12 (95% CIs 0.99, 1.27; I^2^ 51%). For subclinical coronary atherosclerosis, which included 11 cross-sectional studies and 24,820 adults, the pooled OR was 1.44 (95% CIs 1.15, 1.79; I^2^ 88%). (4) Conclusions: The highest serum phosphate concentrations were positively associated with a 44% increased risk of cardiovascular mortality and subclinical coronary atherosclerosis.

## 1. Introduction

Cardiovascular diseases (CVDs) are the leading causes of morbidity and mortality in developed countries despite the improvements in primary prevention and treatments in recent decades [1,2,3,4,5]. Moreover, the Global Burden of Disease study in 2019 [4] showed that the incidence of CVDs has been steadily increasing for decades in almost all non-high-income countries. This burden of disease can be assessed as CVD mortality, CVD incidence, as well as the presence of subclinical coronary atherosclerosis when the disease has not been identified yet. In this sense, the incidence of atherosclerotic CVDs has declined in the past two decades in the adult population but not in the younger segment; therefore, prevention must start at the earliest possible stage [6].

Along with the traditional cardiovascular risk factors, such as dyslipidemia, hypertension, and tobacco consumption [7,8], new non-traditional exposures have been investigated. Some of them are of increasing interest due to their high presence in our habitual diet, mainly when consuming ultra-processed foods (UPFs). Among them, the exposure to phosphates has been of interest in the last 20 years [9,10,11].

Phosphorus is an essential mineral in the human body, necessary for numerous physiological functions, such as bone health, energy metabolism, and signal transduction [12,13,14]. The body obtains this mineral from food. Thus, the main dietary sources of phosphorus in its natural form (as organic phosphorus) are foods high in protein (such as dairy products, legumes, meat, and fish). However, inorganic phosphorus salts are added to food as additives, specifically in UPFs [15]. Common food items with significant amounts of added phosphorus are processed meat (such as ham and sausages), canned fish, baked goods, cola drinks, and other soft drinks [16]. Unlike organic phosphorus, inorganic phosphorus has higher bioavailability, being almost completely absorbed in the gastrointestinal tract [17,18,19,20]. In the human body, almost all phosphorus is combined with oxygen to form phosphate. Phosphate is primarily present in the blood in its organic form, bound to proteins, while the remaining (approximately 30%) is in its ionized form, also called inorganic phosphate, and freely circulates in the bloodstream [21,22].

Previously, only two meta-analyses assessing the association between serum phosphate and cardiovascular risk have been published. One of them examined the association between elevated fasting serum phosphate concentration and the risk of death and CVDs in individuals with chronic kidney disease (CKD) [23]. Therefore, the results could not be generalized to the general population. The second one assessed the association of serum phosphate with all-cause and cardiovascular mortality in the general population; however, it was based on a limited number of studies [24]. The study of this association is of great interest, and recently published articles suggest that even an elevated serum phosphate concentration within the normal range might contribute to increased cardiovascular risk independently of kidney function [25,26].

Our aim was to conduct a meta-analysis based on published epidemiological studies assessing the association between serum phosphorus concentration and cardiovascular mortality in non-selected samples of the general population. CVD incidence and subclinical coronary atherosclerosis were also considered as main outcomes.

## 2. Materials and Methods

### 2.1. Search Strategy and Selection Criteria

From conception of this meta-analysis, we followed the Cochrane Handbook for Systematic Review of Intervention [27], and the search was conducted according to the Preferred Reporting Items for Systematic Review and Meta-Analysis (PRISMA) statement recommendations [28]. The results were reported according to the Meta-analysis of Observational Studies in Epidemiology (MOOSE) [29] and PRISMA reporting guidelines [28]. The review methods were established “a priori” before the review was carried out. The study protocol was registered in the International Prospective Register of Systematic Reviews (PROSPERO): CRD42023433390.

An advanced search of published evidence was conducted to identify articles on the association between serum levels of phosphate and cardiovascular mortality, CVDs, and subclinical coronary atherosclerosis. The data sources included the electronic databases PubMed, SCOPUS, and the Web Of Science from inception to March 2024 (included). The review was carried out using specific keywords related to the outcomes of interest and was conducted using MeSH terms to create a uniform and comprehensive search strategy, for which the terms and language filters are listed in Table 1.

### 2.2. Study Selection

The inclusion criteria were defined according to the following items: (a) studies assessing serum phosphate–cardiovascular mortality, serum phosphate–CVD (coronary heart disease, stroke, and heart failure), and serum phosphate–subclinical coronary atherosclerosis relationships in population-based epidemiological studies (experimental, observational, as well as cross-sectional studies); (b) studies conducted in humans aged over 18 without previously established CVDs; (c) studies reporting Hazard Ratio (HR), Odds Ratio (OR), or Prevalence Ratio (PR) with their corresponding 95% Confidence Interval (CI).

The exclusion criteria were defined according to the following items: (a) pregnant women, because in this population, volume could affect serum phosphate concentrations, making the results non comparable; (b) studies assessing outcomes other than the main ones considered, such as aortic stenosis, left ventricular mass, atrial fibrillation, arterial stiffness, endothelial dysfunction, or using a score to predict cardiovascular risk; (c) studies performed in selected populations (for example, on secondary cardiovascular prevention, diabetics, patients with chronic renal disease, or multiple myeloma); (d) studies with low methodological quality for which the validity of the study results was threatened; and (e) in addition, we excluded reviews, meta-analyses, conference articles, letters to the editor, and articles in which essential information to be meta-analyzed was missing.

### 2.3. Data Extraction and Risk of Bias Assessment

Data extraction was undertaken independently by two reviewers (CT-B and PG-C), who screened articles titles and abstracts. Disagreements between these researchers were discussed with another author (BM-F) and settled by consensus.

After reading the full text and Appendix A of the selected studies, the following data from each article were extracted: the first author’s name, publication year, country of the study (sample), sample size, men (%), age, follow-up time, outcome, diagnostics and measurements, number of cases, association estimates (HR, OR, or PR) when comparing the highest vs. the reference level of serum phosphate concentrations, and adjustment for confounding factors. Finally, the Joanna Briggs Institute (JBI) critical appraisal checklist tool for cohort and case–control studies was selected to assess the risk of bias [30].

### 2.4. Statistical Analysis

We calculated central pooled estimates of the association for the highest category compared to the reference category of serum phosphate concentrations and their 95% CIs. Heterogeneity among studies was assessed by using the I^2^ test. An I^2^ > 30% indicated heterogeneity, and then a random-effects model was used; otherwise, a fixed-effects model was performed. Furthermore, when a study reported data separately for men and women, we entered the data as independent studies. Publication bias was examined by visual inspection of the funnel plot as well as by calculating the Egger’s test (*p* value < 0.05 indicated the presence of publication bias) [31]. Analyses were performed with R statistical software (version 4.0.4).

## 3. Results

We identified 2795 records through our search. After removing replicates, exclusions were made after reading the title or the abstract. A total of 56 reports were sought for retrieval and assessed for eligibility. Of these, 29 articles were excluded because they did not meet the inclusion criteria, and two articles were excluded because other outcomes were evaluated [32,33]. Finally, 25 quality studies were included in the review (all of them were cohort or cross-sectional studies). No articles were excluded due to risk of bias (Figure 1).

A total of seven articles that assessed the relationship between serum phosphate concentrations and cardiovascular mortality were included [22,34,35,36,37,38,39]. For CVDs, eight articles were included [39,40,41,42,43,44,45,46]. Finally, we found 11 articles reporting the relationship between serum phosphate concentrations and subclinical atherosclerosis (independently of the method used for measurement), whether it was reported as coronary arterial calcification (CAC) or as the presence of atheroma plaque [1,14,25,26,47,48,49,50,51,52,53].

### 3.1. Cardiovascular Mortality

Cardiovascular mortality was obtained based on the specific cause of death from national registries, death certificates, or medical records. Seven prospective cohort studies [22,34,35,36,37,38,39] assessed the relationship between serum phosphate concentrations and cardiovascular mortality. All of the studies had a low risk of bias, reaching at least 9 points according to the JBI critical appraisal checklist (Table 2).

The results were meta-analyzed using a fixed-effects model because no evidence of heterogeneity was observed (I^2^ = 0%) after combining the results that compared the highest level of serum phosphate versus the category of reference. A total of 41,764 participants were included. The pooled estimate (HR 1.44, 95% CI 1.28–1.61, *p* = 0.46) showed a 44% increase in cardiovascular mortality risk in participants with elevated serum phosphate concentrations, which was statistically significant (Figure 2). Publication bias was not observed by examination of the funnel plot (Appendix A) nor by using the Egger’s test (*p* = 0.7023).

### 3.2. Cardiovascular Diseases

Information on CVDs was obtained by reviewing hospital records of those participants who mentioned in the interviews having suffered from coronary heart disease, stroke, or heart failure or by reviewing the country’s health databases. Thirteen articles were included in the systematic review, including seven studying the relationship with coronary heart disease [39,40,41,43,45,46], three with stroke [39,43,45], and three with heart failure [40,42,44]. The designs of the selected articles were nine prospective cohort studies and one case–control study. All of the studies had a low risk of bias according to the JBI critical appraisal checklist, reaching at least 10 points for the prospective cohort studies as well as the maximum score for the cross-sectional studies (Table 3).

The JBI criteria for case–control studies include: (1) comparable groups; (2) cases and controls matched appropriately; (3) criteria for identification of cases and controls; (4) exposure measurement; (5) exposure measured in the same way for cases and controls; (6) confounding factors; (7) strategies to deal with confounders; (8) outcome measurement; (9) exposure period of interest sufficient; and (11) statistical analysis. The eight prospective cohort studies that were meta-analyzed [25,41,42,43,44,45,46,47] provided information on coronary heart disease (six studies), stroke (three studies), and heart failure (three studies). The results were combined using a random-effects model as evidence of heterogeneity was observed (I^2^ = 51%), after combining the results that compared the highest level of serum phosphate versus the category of reference. A total of 61,723 participants were included. The pooled estimate (HR 1.12, 95% CI 0.99–1.27, *p* < 0.01) showed a 12% increase in the incidence of CVDs in participants with elevated serum phosphate concentrations, which was marginally significant (Figure 3). Publication bias was not observed by examination of the funnel plot (Appendix A) nor by using the Egger’s test (*p* = 0.3916). 

### 3.3. Subclinical Coronary Atherosclerosis

Subclinical coronary atherosclerosis was assessed by computed tomography in all of the studies. However, subclinical atherosclerosis was measured differently by the presence of at least one plaque or by using different scores (Agatston Score, Gensi Score, and Fresinger Score). Eleven cross-sectional studies [1,14,25,26,47,48,49,50,51,52,53] assessed the relationship between serum phosphate concentrations and subclinical coronary atherosclerosis. All of the studies had a low risk of bias, reaching the maximum score according to the JBI critical appraisal checklist (Table 4).

The results were meta-analyzed using a random-effects model as evidence of heterogeneity was observed (I^2^ = 88%), after combining the results that compared the highest level of serum phosphate versus the category of reference. A total of 24,820 participants were included. The pooled estimate (OR 1.44, 95% CI 1.15–1.79, *p* < 0.01) showed a 44% increase in the occurrence of subclinical coronary atherosclerosis in individuals with elevated serum phosphate concentration, which was statistically significant (Figure 4). Publication bias was observed by examination of the funnel plot (Appendix A) and by using the Egger’s test (*p* = 0.004). 

## 4. Discussion

This updated meta-analysis is based on 25 epidemiological studies, including a total of 126,614 adults from the general population. The results provide important evidence about the consistent positive relationship between elevated serum phosphate concentrations and cardiovascular outcomes. Individuals with higher serum phosphate levels have a 44% higher risk of cardiovascular death. For subclinical coronary atherosclerosis among participants previously free of CVDs, the results are similar. Thus, individuals with higher serum phosphate levels also have a 44% higher prevalence of subclinical coronary atherosclerosis. For CVD events, the association is also positive, but marginally significant. As far as we know, this is the first meta-analysis that shows the relationships between serum phosphate and CVD events as well as subclinical coronary atherosclerosis.

For cardiovascular mortality, our findings are consistent with a previous review [24] suggesting that individuals with higher serum phosphate levels had a 36% higher risk of cardiovascular mortality. In this updated meta-analysis that comprises seven high-quality studies, the association found between serum phosphate and cardiovascular mortality is slightly higher, and no heterogeneity was observed among studies. In the same way, no evidence of publication bias was observed, suggesting that the observed association is valid. This meta-analysis provides the best available evidence to assess the studied associations. However, we do not have enough evidence to draw conclusions about the variation in associations by sex.

For cardiovascular events, 12 cohort studies were included in this meta-analysis. Our results show a positive association without achieving statistical significance. High heterogeneity was found, probably due to the inclusion of three different outcomes (coronary heart disease, stroke, and heart failure). The number of studies was not enough to carry out separate analyses. Therefore, more evidence is needed for this outcome.

For subclinical coronary atherosclerosis, 11 studies were selected. High heterogeneity across studies was found, probably due to the different definitions used to characterize subclinical coronary atherosclerosis (presence of plaques or different calcium scores). It is of note that all of the studies had a cross-sectional design. However, since subclinical coronary atherosclerosis is asymptomatic, participants did not modify their behaviors (including dietary habits) because of the presence of the disease. We also highlight that all of the articles had a high-quality score and low risk of bias. However, in this analysis, we could not rule out the presence of publication bias.

Dietary phosphorus derived from inorganic sources may have a greater influence on serum phosphate [54]. High absolute phosphorus intake has been associated with all-cause mortality starting from 1400 milligrams of phosphorus per day, which is twice the U.S. Recommended Daily Allowance for adults. Also, it has been reported that more than one-third of Americans consume more than 1400 mg of phosphorus per day [55]. Phosphorus additives are often added to UPFs during their processing to improve certain product characteristics, such as flavor, creaminess, preservation, and juiciness, or to prevent separation of beverages into individual ingredients [17]. Currently, there is considerable concern about the increased consumption of UPFs in recent decades [56,57], as a UPF-rich diet increases daily phosphorus intake by 250–1000 mg compared with a diet based on fresh and unprocessed foods [58]. Therefore, effective algorithms have been proposed to identify children at intermediate-high risk of future CVDs in order to promote the development of preventive programs that may personalize healthy lifestyles, behavioral modifications, and improved nutritional habits [59].

Serum phosphate levels are regulated by several complex compensatory mechanisms, including gastrointestinal absorption, intracellular displacement, and renal excretion [60,61]. Intestinal absorption of phosphorus depends on its form and source. Natural organic phosphorus sources are less digested and therefore less bioavailable, with absorption from 20 to 60% [62]. By contrast, inorganic phosphorus is more bioavailable, being almost completely absorbed in the digestive tract, as it does not require enzymatic digestion and dissociates rapidly in stomach acid; phosphorus additives are composed of inorganic phosphate [15,17,18,19,20]. In addition, it is possible that dietary phosphorus derived from inorganic sources may have a greater influence on serum phosphate and parathyroid hormones [62].

In healthy subjects, serum phosphate is almost completely filtered through the renal glomerulus, and 80–90% is reabsorbed. Total renal phosphate excretion is balanced with phosphorous intake [20,63]. Therefore, the measurement of urinary phosphate excretion in individuals with preserved renal function is considered a reliable marker of intestinal phosphorus absorption.

Regarding CVDs and serum phosphate, both postprandial elevation of serum phosphate and continued elevation of serum phosphate should be considered in the development of CVDs [64]. Certain in vitro studies have shown that hyperphosphatemia induces phenotypic conversion of vascular smooth muscle cells into osteoblast-like cells that express biochemical markers characteristic of bone tissue, resulting in vascular calcification in humans [65]. In addition, both in vitro and in vivo studies have demonstrated that high dietary phosphorus intake causes endothelial dysfunction within a short period, suggesting that the elevation of serum phosphate concentrations due to dietary phosphorus load may be a risk factor for endothelial dysfunction in both healthy individuals and those with CKD [64]. However, the causal relationship between higher serum phosphate concentrations and coronary atherosclerosis has not been fully explained yet.

Our study has some strengths. First, we explore separately the association between serum levels of phosphate and the risk of mortality, CVDs, and subclinical coronary atherosclerosis in primary prevention. Therefore, for CVD incidence and subclinical coronary atherosclerosis, studies involving individuals with previous CVDs were excluded. In addition, the included studies in these meta-analyses had high methodological quality. In cohort studies as well as in cross-sectional studies, the results were adjusted for an important number of confounding factors. Finally, the PRISMA guidelines were followed in performing the meta-analysis and reporting the results.

However, our study also has some possible limitations. First, the results show a high degree of heterogeneity in two of the study outcomes (I^2^ = 51% for CVDs and I^2^ = 88% for subclinical coronary atherosclerosis). This heterogeneity and the presence of a certain degree of publication bias for subclinical coronary atherosclerosis could be related to differences in the measurement of serum phosphate as well as the quantification of subclinical coronary atherosclerosis, as different measurements were used. Moreover, all CVDs were meta-analyzed together, without being able to distinguish among different types. Also, in all cases, serum phosphate concentrations were measured at a single time point, without considering changes over time, which could lead to non-differential misclassification. Similarly, changes in weight loss [66] or changes in other confounding factors were not taken into account in the published articles, which could have led to regression dilution bias.

## 5. Conclusions

Higher serum phosphate concentrations are positively associated with a 44% increase in mortality and subclinical coronary atherosclerosis in the general population. In the future, it is desirable to publish more studies that evaluate the association between serum phosphate and incident CVDs in order to independently assess the association for coronary heart disease, stroke, and heart failure.

## Figures and Tables

**Figure 1 nutrients-16-01599-f001:**
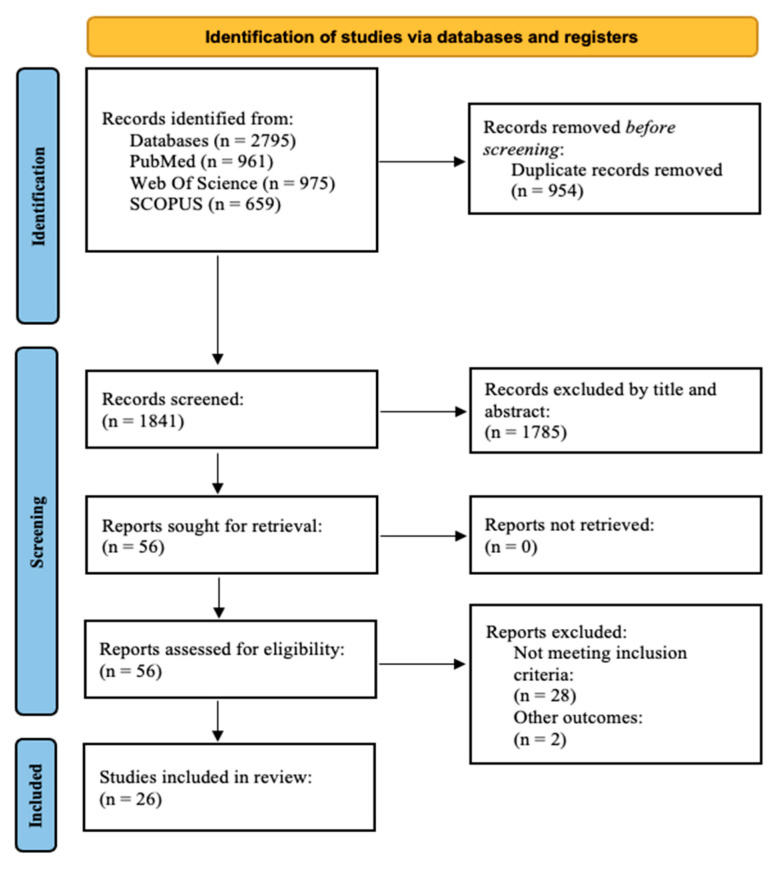
PRISMA 2020 flow diagram for systematic reviews.

**Figure 2 nutrients-16-01599-f002:**
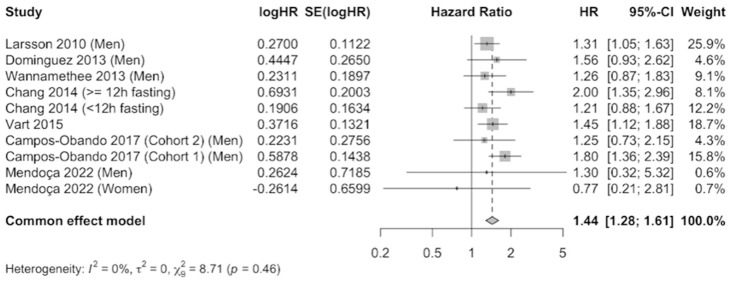
Forest plot of association of serum phosphate with cardiovascular mortality using a fixed-effects model [22,34,35,36,37,38,39].

**Figure 3 nutrients-16-01599-f003:**
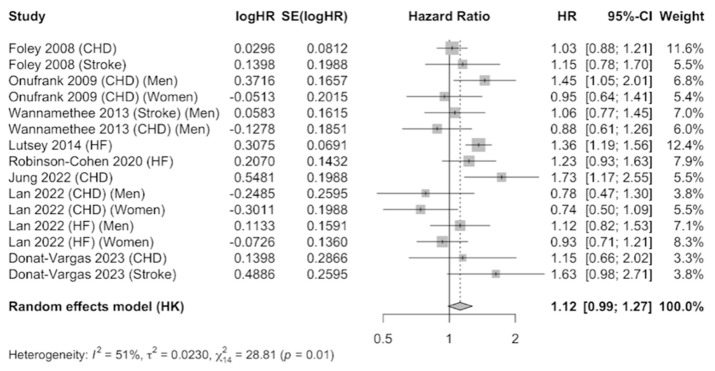
Forest plot of association of serum phosphate with cardiovascular disease using a random-effects model [39,40,41,42,43,44,45,46].

**Figure 4 nutrients-16-01599-f004:**
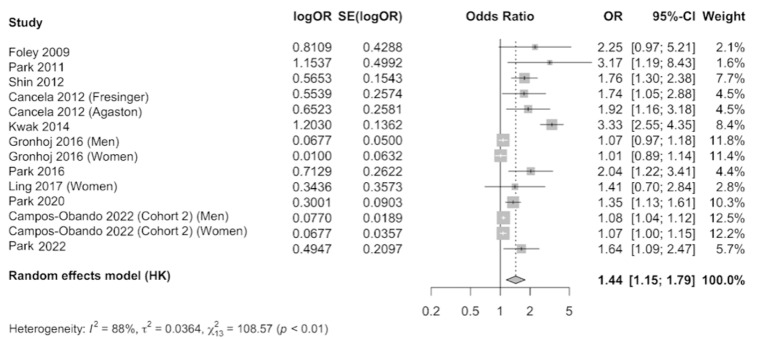
Forest plot of association of serum phosphate with subclinical coronary atherosclerosis using a random-effects model [1,14,25,26,47,48,49,50,51,52,53].

**Table 1 nutrients-16-01599-t001:** Search strategy in selected databases.

**PubMed**	((“Phosphates/blood” [Mesh] OR “Phosphorus/blood” [Mesh]) AND (“Cardiovascular Mortality” OR “Cardiovascular Diseases” [Mesh] OR “Stroke” [Mesh] OR “Atherosclerosis” [Mesh] OR “Coronary Artery Disease” [Mesh] OR “Carotid Artery Diseases” [Mesh] OR “Coronary Disease” [Mesh] OR “Femoral Artery Diseases”)) AND ((English [Filter] OR Spanish [Filter]) AND (Humans [Filter])).
**SCOPUS**	TITLE-ABS-KEY (“Serum Phosphate” OR “Serum Phosphorus” OR “Plasma Phosphate” OR “Plasma Phosphorus”) AND TITLE-ABS-KEY (“Cardiovascular Mortality” OR “Cardiovascular Disease” OR “Stroke” OR “Atherosclerosis” OR “Coronary Artery Disease” OR “Carotid Artery Disease” OR “Coronary Disease” OR “Femoral Artery Disease”) AND LANGUAGE (English) OR LANGUAGE (Spanish) AND (LIMIT-TO (SRCTYPE, “j”)) AND (LIMIT-TO (DOCTYPE, “ar”) OR LIMIT-TO (DOCTYPE, “re”)) AND (LIMIT-TO (LANGUAGE, “English”) OR LIMIT TO (LANGUAGE, “Spanish”)).
**Web Of Science**	(“Serum Phosphate” OR “Serum Phosphorus” OR “Plasma Phosphate” OR “Plasma Phosphorus” (Topic)) AND (“Cardiovascular Mortality” OR “Cardiovascular Disease” OR “Stroke” OR “Atherosclerosis” OR “Coronary Artery Disease” OR “ Carotid Artery Disease” OR “Coronary Disease” OR “Femoral Artery Disease” (Topic)) AND (English OR Spanish (Language)) AND (Article OR Review Article (Document Types)).

**Table 2 nutrients-16-01599-t002:** Characteristics of the cardiovascular mortality studies sorted by year of publication.

Author (Year)	Country (Sample)	Sample Size	Men (%)	Age	Follow-Up	Nº of Cases	Results	Adjustment for Confounders	Quality Score (JBI Checklist)
Mendoça et al. (2022) [38]	U.S.A. (NHANES 2003–2006)	5698	44%	≥18 y	81 months	N/A (n total = 141)	HR (95% CI) (T2 vs. T3) Men: 1.30 (0.32– 5.35) Women: 0.77 (0.21– 2.79)	Age, sex, poverty–income ratio, education, BMI, eGRF rate, albumin–creatinine ratio, non–HDL cholesterol, C-reactive protein, albumin, protein intake, HTA, diabetes, history of MI, history of stroke and smoking status, PTH, 25-hydroxyvitamin D, serum Ca, dietary Ca, and dietary phosphorus.	9/11 (JBI: 9, 10) Included
Campos- Obando et al. (2017) [22]	The Netherlands (Rotterdam Study Cohort I and II (RS I and II)	2710 RS-I: 1577 RS-II: 1133	100%	≥55 y	RS-I: 22 y RS-II: 12 y	RS-I: 266 RS-II: 77	HR (95% CI) per 1 mg/dL increase in P levels. RS-I: 1.80 (1.35–2.39) RS-II: 1.25 (0.73– 2.15)	Age, BMI, and smoking.	9/11 (JBI: 9, 10) Included
Vart et al. (2015) [37]	U.S.A. (NHANES 1988–1994)	15,833	N/A	≥18 y	14.3 y	1691	HR (95% CI) (<4.2 vs. ≥4.2 mg/dL) 1.45 (1.12–1.88)	Age, gender, race, and eGRF rate.	10/11 (JBI: 10) Included
Chang et al. (2014) [35]	U.S.A. (NHANES 1988–1994)	12,984 <12 h fasting: 6312 ≥12 h fasting: 6633	<12 h fasting: 47.5% ≥12 h fasting: 48.2%	≥20 y	14.3 y	N/A	HR (95% CI) (Q4 vs. Q1) <12 h fasting: 1.21 (0.88–1.67) ≥12 h fasting: 2.00 (1.36–2.96)	Examination session (morning vs. afternoon/evening), age, sex, race, ethnicity, poverty–income ratio, body mass index, systolic blood pressure, diabetes, smoking status, physical activity, non-HDL cholesterol level, log albumin–creatinine ratio, eGRF rate, and vitamin D status.	10/11 (JBI: 10) Included
Dominguez et al. (2013) [34]	U.S.A. Osteoporotic Fractures in Men (MrOS) Study	670	100%	≥65 y	9.3 y	N/A	HR (95% CI) (Q4 vs. Q1) 1.56 (0.93–2.62)	Age and race, eGRF rate, microalbuminuria (yes/no), prevalent CVD, diabetes, systolic blood pressure, blood pressure medication use, tobacco use (current, former, never), BMI, total cholesterol level, HDL cholesterol level, and lipid medication use.	11/11 Included
Wannamethee et al. (2013) [39]	England (British Regional Heart Study)	1693	100%	60–79 y	11 y	166	HR (95% CI) (Q4 vs. Q1) 1.26 (0.87–1.83)	Age, cigarette smoking, alcohol intake, physical activity, social class, BMI, use of antihypertensive drugs, diabetes function, systolic blood pressure, eGRF rate, C-reactive protein, and von Willebrand factor.	10/11 (JBI: 10 Included
Larsson et al. (2010) [36]	Sweden (Rotterdam Study)	2176	100%	50 y	29.8 y	466	HR (95% CI) (Q4 vs. Q1) 1.31 (1.06–1.63)	Age, albumin, eGRF rate, diabetes, use of antihypertensive medication, systolic and diastolic blood pressures, total cholesterol, triglycerides, BMI, and smoking.	9/11 (JBI: 1, 10) Included

BMI: Body Mass Index; Ca: Calcium; CI: Confidence Interval; CVD: Cardiovascular Disease; eGRF: Estimated Glomerular Filtration; HDL cholesterol: High Density Lipoprotein Cholesterol; HR: Hazard Ratio; HTA: Hypertension; MI: Myocardial Infarction; P: Phosphate; PTH: Parathyroid Hormone. JBI criteria for cohort studies: (1) similar groups and from the same population; (2) exposure measured similarly in exposed and unexposed groups; (3) exposure measurement; (4) confounding factors; (5) strategies to deal with confounders; (6) free of the outcome at the start of the study; (7) outcome measurement; (8) follow-up time reported and sufficient; (9) losses to follow-up; (10) strategies to address incomplete follow-up; and (11) statistical analysis.

**Table 3 nutrients-16-01599-t003:** Characteristics of the cardiovascular disease studies sorted by year of publication.

Author (Year)	Country (Sample)	Study Design	Sample Size	Men (%)	Age	Follow-Up	Outcome	Nº of Cases	Results	Adjustment for Confounders	Quality Score (JBI Checklist)
Donat-Vargas et al. (2023) [45]	Sweden (Swedish Mammography Cohort 1987–1990)	Cohort study	1625	0%	56–85 y	9.4 y	CHD	75	HR (95% CI) (T1 vs. T3) 1.15 (0.66–2.03)	Age, BMI, education, family history of MI < 60 years, history of diabetes, history of HTA, smoking, walking/cycling >20 min/day, leisure time inactivity >5 h/day, adherence to Mediterranean diet, alcohol consumption, vitamin D supplement use, eGRF rate, urinary sodium (mmol/mmol creatinine), plasma Ca (mmol/L), use of diuretics (ATC-codes C03), and plasma P (mmol/L).	11/11 Included
Donat-Vargas et al. (2023) [45]	Sweden (Swedish Mammography Cohort 1987–1990)	Cohort study	1625	0%	56–85 y	9.4 y	STROKE	101	HR (95% CI) (T1 vs. T3) 1.63 (0.98–2.71)	Age, BMI, education, family history of MI < 60 years, history of diabetes, history of HTA, smoking, walking/cycling >20 min/day, leisure time inactivity >5 h/day, adherence to Mediterranean diet, alcohol consumption, vitamin D supplement use, eGRF rate, urinary sodium (mmol/mmol creatinine), plasma Ca (mmol/L), use of diuretics (ATC-codes C03), and plasma P (mmol/L).	11/11 Included
Jung et al. (2022) [41]	South Korea (Health Risk Assessment study, and Health Insurance Review and Assessment Service)	Cohort study	15,259	52.3%	30–85 y	50 months	CHD	315	HR (95% CI) (Q1 vs. Q4) 1.73 (1.18–2.55)	Age, sex, BMI, smoking status, alcohol intake, physical activity, mean arterial blood pressure, C-reactive protein level, chronic kidney disease, serum Ca, serum potassium, and eGRF rate.	10/11 (JBI: 9) Included
Lan et al. (2022) [40]	China (Gaohang community)	Cohort study	3948	44.2%	60–79 y	4 y	CHD	371	HR (95% CI) (Q1 vs. Q4) Men: 0.78 (0.47–1.30) Women: 0.74 (0.50–1.09)	Systolic blood pressure, total Ca, bicarbonate, fasting glucose, HbA1c, and smoking habits in males; age, systolic blood pressure, waist circumference, BMI, total Ca, bicarbonate, total cholesterol, education, and kidney disease history in females.	11/11 Included
Lan et al. (2022) [40]	China (Gaohang community)	Cohort study	3948	44.1%	60–79 y	4 y	HEART FAILURE	811	HR (95% CI) (Q1 vs. Q4) Men: 1.12 (0.81–1.53) Women: 0.93 (0.71–1.21)	Systolic blood pressure, total Ca, bicarbonate, fasting glucose, HbA1c and smoking habits in males; age, systolic blood pressure, waist circumference, BMI, total Ca, bicarbonate, total cholesterol, education, and kidney disease history in females.	11/11 Included
Robinson- Cohen et al. (2020) [42]	U.S.A. (MESA 2000–2002)	Cohort study	6413	46.8%	45–84 y	14.9 y	HEART FAILURE	333	HR (95% CI) (≤4 vs. >4 mg/dL) 1.13 (0.92–1.63)	Age, sex, gross family income in the past 12 months, educational attainment, in analyses including all participants, ethnicity, BMI, systolic blood pressure, use of antihypertensive medication, low-density total cholesterol, diabetes status, smoking status, and eGRF rate.	10/11 (JBI: 10 Included
Lutsey et al. (2014) [44]	U.S.A. (ARIC 1987–1989)	Cohort study	14,709	45.5%	45–84 y	21.8 y	HEART FAILURE	2250	HR (95% CI) (Q1 vs. Q5) 1.36 (1.18–1.56)	Age, sex, race, center, education, physical activity, smoking status, BMI, prevalent diabetes, systolic blood pressure, HTA medication use, lipid-lowering medication use, prevalent CHD, eGRF rate, LDL cholesterol, HDL cholesterol, triglycerides, and incident CHD as a time-varying covariate.	10/11 (JBI: 10) Included
Wannamethee et al. (2013) [39]	England (British Regional Heart Study 1978–1980)	Cohort study	1693	100%	60–79 y	11 y	CHD	313	HR (95% CI) (Q1 vs. Q4) 0.88 (0.61–1.13)	Age, cigarette smoking, alcohol intake, physical activity, social class, BMI, use of antihypertensive drugs, diabetes function, systolic blood pressure, eGRF rate, C-reactive protein, and von Willebrand factor.	10/11 (JBI: 10) Included
Wannamethee et al. (2013) [39]	England (British Regional Heart Study 1978–1980)	Cohort study	1693	100%	60–79 y	11 y	STROKE	228	HR (95% CI) (Q1 vs. Q4) 1.06 (0.78–1.26)	Age, cigarette smoking, alcohol intake, physical activity, social class, BMI, use of antihypertensive drugs, diabetes function, systolic blood pressure, eGRF rate, C-reactive protein, and von Willebrand factor.	10/11 (JBI: 10) Included
Onufrak et al. (2009) [46]	U.S.A. (ARIC 1987–1989)	Cohort study	13,998	46.8%	45–84 y	13 y	CHD	922	HR (95% CI) (Q2 vs. Q5) Men:1.45 (1.04–2.01) Women: 0.95 (0.63–1.41)	Age, black race, current smoking, diabetes mellitus, HTA, total cholesterol, HDL cholesterol, eGRF rate, current use of estrogen replacement therapy, and menopausal status.	10/11 (JBI: 10) Included
Foley et al. (2008) [43]	U.S.A. (ARIC 1987–1989)	Cohort study	13,616	44.8%	30–75 y	12.6 y	CHD	1666	HR (95% CI) (Q1 vs. Q5) 1.03 (0.88–1.21)	Age, sex, current smoking cigarette smoking years, BMI, HDL cholesterol, LDL cholesterol, triglycerides, serum albumin, eGRF rate, caloric intake, and phosphorus intake.	10/11 (JBI: 10) Included

BMI: Body Mass Index; Ca: Calcium; CHD: Cardiovascular Heart Disease; CI: Confidence Interval; eGRF: Estimated Glomerular Filtration; HbA1C: Glycated Hemoglobin; HDL cholesterol: High Density Lipoprotein Cholesterol; HR: Hazard Ratio; HTA: hypertension; LDL cholesterol: Low Density Lipoprotein Cholesterol; MI: Myocardial Infarction; P: Phosphate. JBI criteria for cohort studies: (1) similar groups and from the same population; (2) exposure measured similarly in exposed and unexposed groups; (3) exposure measurement; (4) confounding factors; (5) strategies to deal with confounders; (6) free of the outcome at the start of the study; (7) outcome measurement; (8) follow-up time reported and sufficient; (9) losses to follow-up; (10) strategies to address incomplete follow-up; and (11) statistical analysis.

**Table 4 nutrients-16-01599-t004:** Characteristics of the subclinical coronary atherosclerosis studies sorted by year of publication.

Author (Year)	Country (Sample)	Sample Size	Men (%)	Age	Diagnostic and Measurement	Nº of Cases	Results	Adjustment for Confounders	Quality Score (JBI Checklist)
Campos-Obando et al. (2022) [26]	The Netherlands (Rotterdam Study Cohort I, II, III and IV (RS I, II, III and IV))	1889	44%	≥18 y	Coronary computed tomography. Agatston score CACs > 100–300	Men: 600/878 Women: 418/1011	PR (95% CI) (Q1 vs. Q5) Men: 1.08 (1.04–1.12) Women: 1.07 (1.00–1.15)	Age, BMI, blood pleasure, smoking, prevalent CVD, prevalent diabetes mellitus, prevalent serum levels of 25-hydroxyvitamin D, total Ca, C-reactive protein, total cholesterol to HDL cholesterol ratio, and glucose.	8/8 Included
Park et al. (2022) [25]	South Korea (Ulsan University Hospital 2014–2020)	1636	43.5%	≥55 y	Coronary computed tomography Any atherosclerosis plaque.	297	OR (95% CI) (T1 vs. T3) 1.64 (1.09–2.48)	Age, sex, systolic blood pressure, diastolic blood pressure, BMI, fasting blood glucose, HbA1C, LDL cholesterol, HDL cholesterol, triglycerides, creatine, and Ca.	8/8 Included
Park et al. (2020) [52]	South Korea (Health Screening and Promotion Center in the Asan Medical Center 2007–2011)	6329	72.9%	≥18 y	Coronary computed tomography Any atherosclerosis plaque.	2634	OR (95% CI) (Q1 vs. Q4) 1.35 (1.13–1.61)	Age, sex, obesity, current smoking, HTA, diabetes mellitus, hyperlipidemia, family history of coronary artery disease, high-sensitivity C-reactive protein ≥ 2 mg/dL, and corrected Ca concentration.	8/8 Included
Ling et al. (2017) [14]	China (Cardiology Department of Zhongshan Hospital in Shanghai 2015–2016)	227	0%	≥20 y	Coronary computed tomography Gensi score Gensi score > 0	111	OR (95% CI) (≤3.59 vs. >3.59 mg/dL) 1.41 (0.70–2.84)	Age, BMI, smoking, HTA, diabetes, LDL cholesterol, HDL cholesterol, triglycerides, eGRF rate, statin use, and high-sensitivity C-reactive protein.	8/8 Included
Grønhøj et al. (2016) [1]	Denmark (Danrisk Study 2009–2010)	1046	47%	≥65 y	Coronary computed tomography. Agatston score CACs divided in 4 categories: 0, 1–99, 100–399, ≥400	1-99 AU: 335 (M 196/W 139) 100-399 AU: 98 (M 66/W 32) ≥400 AU: 54 (M 41/W13)	OR (95% CI) per 0.31 mg/dL increase in P levels. Men: 1.07 (0.97–1.18) Women: 1.01 (0.90–1.14)	Creatine, age, gender, smoking, HTA, hypercholesterolemia, and family history of CVDs.	8/8 Included
Park et al. (2016) [48]	South Korea (Ulsan University Hospital 2009–2013)	2509	63.2%	60–79 y	Coronary computed tomography Agatston score CACs > 100	307	OR (95% CI) (Q4 vs. Q1) 2.04 (1.22–3.41)	Age, sex, diabetes, BMI, systolic blood pressure, corrected serum Ca, albumin, HbA1c, LDL cholesterol, and HDL cholesterol.	8/8 Included
Kwak et al. (2014) [53]	South Korea (Kangbuk Samsung Health Study)	23,652	83.5%	49.8 ± 7.3 y	Coronary computed tomography Agatston score CACs ≥ 100	437	OR (95% CI) (Q4 vs. Q1) 3.33 (2.55–4.35)	Age, sex, smoking status, alcohol intake, physical activity, BMI, educational level, family history of CVDs, medication for dyslipidemia, diabetes mellitus, HTA, eGRF rate, albumin, ferritin, total calorie intake, Ca intake, phosphorus intake, and Ca supplements.	8/8 Included
Shin et al. (2012) [50]	South Korea (Severance Hospital 2004–2009)	7553	57%	≥30 y	Coronary computed tomography Agatston score CACs > 100	8%	OR (95% CI) (Q4 vs. Q1) 1.76 (1.29–2.38)	Age, male gender, BMI, HTA, diabetes, smoking, dyslipidemia, and proteinuria.	8/8 Included
Cancela et al. (2012) [47]	Brasil (Heart Institute of Hospital das Clínicas 2008–2009)	290	57.5%	≥18 y	Coronary computed tomography Agatston score CACs > 10	169	OR (95% CI) (T3 vs. T1) 1.92 (1.56–3.19)	Age, gender, diabetes, HTA, FGF23 level, and PTH concentration.	8/8 Included
Cancela et al. (2012) [47]	Brasil (Heart Institute of Hospital das Clínicas 2008–2009)	286	N/A	≥18 y	Coronary computed tomography. Fresinger score Fresinger score > 4	137	OR (95% CI) (T3 vs. T1) 1.74 (1.06–2.88)	Gender, race, age, HTA, diabetes, logPTH, and logFGF23.	8/8 Included
Park et al. (2011) [49]	South Korea (Asan Medical Center 2007–2009)	402	64%	≥18 y	Coronary computed tomography Agatston score CACs > 100	75	OR (95% CI) (Q4 vs. Q1) 3.17 (1.19–8.41)	Age, sex, BMI, diabetes, HTA, systolic and diastolic blood pressures, family history of CVDs, corrected serum Ca, Ca-phosphorus product, serum glucose, total, HDL cholesterol, and triglycerides.	8/8 Included
Foley et al. (2009) [51]	U.S.A. (CARDIA 1985–1986)	3015	44.6%	18–30 y	Coronary computed tomography Agatston score CACs ≥ 100 vs. <100	1.6%	OR (95% CI) (Q4 vs. Q1) 2.25 (0.97–5.21)	Age, gender, race, education, BMI cigarette, HTA, diabetes, exercise, glucose, family history of MI, LDL cholesterol, HDL cholesterol, triglycerides, systolic and diastolic blood pressures, phosphorus intake, Ca intake, fat intake, alcohol intake, protein intake, carbohydrate intake, and medications.	8/8 Included

AU: Angaston Units; BMI: Body Mass Index; Ca: Calcium; CACS: Coronary Artery Calcium Score; CI: Confidence Interval; CVD: Cardiovascular Disease; eGRF: Estimated Glomerular Filtration; FGF23: Fibroblast Growth Factor 23; HbA1C: Glycated Hemoglobin; HDL cholesterol: High Density Lipoprotein Cholesterol; HTA: hypertension; LDL cholesterol: Low Density Lipoprotein Cholesterol; MI: Myocardial Infarction; OR: Odds Ratio; PR: Prevalence Ratio; PTH: Parathyroid Hormone. JBI criteria for cross-sectional studies: (1) criteria for inclusion; (2) the study subjects and the setting; (3) exposure measurement; (4) standard criteria used for measurement; (5) confounding factors; (6) strategies to deal with confounders; (7) outcome measurement; (8) statistical analysis.

## Data Availability

Data described in the manuscript, code book, and analytic code will be made available upon request from the corresponding author.

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
