# Peer review of "High Serum Phosphate Is Associated with Cardiovascular Mortality and Subclinical Coronary Atherosclerosis: Systematic Review and Meta-Analysis"

_nutrients, 2024, doi:10.3390/nu16111599_

Round 1
Reviewer 1 Report
Comments and Suggestions for Authors
Page 2 Lines 46-47 Introduction "...the incidence of CVD has been steadily increasing for decades in almost all non-high-income countries".
There is a more dramatic evidence, which could be included in the discussion, if not in the introduction, to emphasize the evidence you have reported and to promote improved nutritional habits since childhood.
:"The incidence of atherosclerotic cardiovascular disease (CVD) has declined in the past 2 decades in the adult population but not in the younger segment (premature CVD) [1Vikulova DN, et al. Premature Atherosclerotic Cardiovascular Disease: Trends in Incidence, risk Factors, and Sex Related Differences, 2000 to 2016. J Am Heart Assoc 2019 :e012178.].
While there is no universal consensus, CVD is currently defined as “premature” if the age at onset is <55 years in males and <65 years in females.
Effective algorithms to identify children at intermediate-high risk of future CV events have been proposed to promote the development of preventive programs which may personalize healthy lifestyles, behavioral modifications, and improved nutritional habits. [Reference:Ferraro S et al.Prediction of atherosclerotic cardiovascular risk in early childhood, Clinica Chimica Acta, Volume 552, 2024.] Your evidence could be considered for improving behavioural habits.
Second point: In each study you should add median and percentile levels of phosphate in each case series, further reporting reference intervals and laboratory method.
Third point: if possible consider what (%?) increase of Phosphate is associated to adverse outcomes or what range of values are associated to adverse outcomes.
In your paper you report high levels of phosphate, anyway we have no idea of what range of levels have to be considered as a risk factor for the different adverse outcomes.
Comments on the Quality of English LanguageEnglish editing required
Author Response
Thank you for your attention, revision, and your suggestions.
Comments on the manuscript.
1. Page 2 Lines 46-47 Introduction "...the incidence of CVD has been steadily increasing for decades in almost all non-high-income countries".
There is a more dramatic evidence, which could be included in the discussion, if not in the introduction, to emphasize the evidence you have reported and to promote improved nutritional habits since childhood: "The incidence of atherosclerotic cardiovascular disease (CVD) has declined in the past 2 decades in the adult population but not in the younger segment (premature CVD) [1Vikulova DN, et al. Premature Atherosclerotic Cardiovascular Disease: Trends in Incidence, risk Factors, and Sex Related Differences, 2000 to 2016. J Am Heart Assoc 2019 :e012178.].
While there is no universal consensus, CVD is currently defined as “premature” if the age at onset is <55 years in males and <65 years in females. Effective algorithms to identify children at intermediate-high risk of future CV events have been proposed to promote the development of preventive programs which may personalize healthy lifestyles, behavioral modifications, and improved nutritional habits. [Reference: Ferraro S et al. Prediction of atherosclerotic cardiovascular risk in early childhood, Clinica Chimica Acta, Volume 552, 2024.] Your evidence could be considered for improving behavioural habits.
Thank you for your suggestion. We have included the following sentence in order to highlight importance of primary prevention early in the life course:
“In this sense, the incidence of atherosclerotic CVD has declined in the past 2 decades in the adult population but not in the younger segment, therefore, prevention must start at the earliest possible stage [6]”. (Lines 49-51).
2. In each study you should add median and percentile levels of phosphate in each case series, further reporting reference intervals and laboratory method.
Thank you for your comment. Following your suggestion, we have attached a table with laboratory methods and reference interval, that could be added as supplementary material.
Table 1. Characteristics of the cardiovascular mortality studies sorted by year of publication.
|
|
Laboratory method |
Reference intervals |
|
Mendoça (2022) |
Ammonium molybdate |
Tertile 2 is considered as the reference. - T1 (M <3.6 mg/dL / W <3.8 mg/dL) - T2 (M 3.6-4.1 mg/dL / W 3.8-4.1 mg/dL) - T3 (>4.1mg/dL) |
|
Campos- Obando (2017) |
Ammonium phosphomolybdate |
Expressed per 1 mg/dL increase in phosphate levels. |
|
Vart (2015) |
Hitachi model 737 multichannel analyzer (Boehringer Mannheim Diagnostics) |
< 4.2 mg/dL (n = 1546/14508) vs ≥ 4.2 mg/dL (n = 141/1325) |
|
Chang (2014) |
Hitachi 7600-110 Chemistry Autoanalyzer (Hitachi, Tokyo, Japan). |
Quartil 1 is considered as the reference. Not available concentration of each quartile. |
|
Dominguez (2013) |
Standard clinical automated analyzer of Roche |
Quartil 1 is considered as the reference. - Q1 (< 3.0 mg/dL) - Q2 (3-3.2 mg/dL) - Q3 (3.3-3.5 mg/dL - Q4 (≥ 3.6 mg/dL) |
|
Wannamethee (2013) |
No data |
Quartil 1 is considered as the reference. - Q1 (< 3.27 mg/dL) - Q2 (3.27-3.57 mg/dL) - Q3 (3.58-3.9 mg/dL) - Q4 (>3.9mg/dL) |
|
Larsson (2010) |
Ammonium molybdate |
Tertile 1 is considered as the reference. - T1 (0.62-2.48 mg/dL) - T2 (2.48-2.79 mg/dL) - T3 (2.79-15.18 mg/dL) |
Table 2. Characteristics of the subclinical coronary atherosclerosis studies sorted by year of publication.
|
|
Laboratory method |
Reference intervals |
|
Donat-Vargas (2023) |
Architect C16000 (Abbott Laboratories, Abbott Park, IL, USA) |
Tertile 1 is considered as the reference. - T1 (2.63-3.35 mg/dL) - T2 (3.38-3.69 mg/dL) - T3 (3.7-4.4 mg/dL) |
|
Jung (2022) |
Hitachi 7600-110 Chemistry Autoanalyzer (Hitachi, Tokyo, Japan). |
Quartil 1 is considered as the reference. - Q1 (≤ 3.4 mg/dL) - Q2 (3.41-3.70 mg/dL) - Q3 (3.71-4.40 mg/dL) - Q4(≥ 4.41 mg/dL) |
|
Lan (2022) |
No data ïƒ standarized methods. |
Quartil 1 is considered as the reference. - Q1 (M < 2.82 mg/dL / W <3.31 mg/dL) - Q2 (M 2.82-3.13 mg/dL / W 3.31-3.59 mg/dL) - Q3 (M 3.13-3.44 mg/dL / W 3.59-3.87 mmol/L) - Q4 (M >3.44 mg/dL / W >3.87 mg/dL) |
|
Robinson- Cohen (2020) |
Timed-rate colorimetry reaction |
>4 vs ≤ 4 mg/dL |
|
Lutsey (2014) |
Ammonium molybdate |
Quintil 1 is considered as the reference. - Q1 (1-3 mg/dL) - Q2 (3.1-3.2 mg/dL) - Q3 (3.3-3.5 mg/dL) - Q4 (3.6-3.8mg/dL) - Q5 (3.9-9.1 mg/dL) |
|
Wannamethee (2013) |
No data |
Quartil 1 is considered as the reference. - Q1 (< 3.27 mg/dL) - Q2 (3.27-3.58 mg/dL) - Q3 (3.58-3.9 mg/dL) - Q4 (>3.88mg/dL) |
|
Taylor (2011) |
Hitachi 917 analyzer using Roche reagents (Roche Diagnostics, Indianapolis, IN) |
Quartil 1 considered as reference). - Q1 (median 2.3 mg/dL) - Q2 (median 2.7 mg/dL) - Q3 (median 3.0 mg/dL) - Q4 (median 3.4 mg/dL) |
|
Onufrak (2009) |
DART phosphorus reagent (Coulter Diagnostics, Hialeah, Florida). |
Quintil 2 is considered as reference). - Q1 (< 3.1 mg/dL) - Q2 (3.1-3.2 mg/dL) - Q3 (3.3-3.5 mg/dL) - Q4 (3.6-3.8 mg/dL) - Q5 (>3.8 mg/dL) |
|
Foley (2008) |
Ammonium molybdate |
Quintil 1 is considered as the reference. - Q1 (< 3.0 mg/dL) - Q2 (3.0-<3.2 mg/dL) - Q3 (3.2-<3.5 mg/dL) - Q4 (3.5-<3.8 mg/dL) - Q5 (≥ 3.8 mg/dL) |
Table 3. Characteristics of the subclinical coronary atherosclerosis studies sorted by year of publication.
|
|
Laboratory method |
Reference intervals |
|
Campos-Obando (2022) |
Ammonium phosphomolybdate |
Quintil 1 is considered as the reference. - Q1 (M 2.57 mg/dL / W 3.04 mg/dL) - Q2 (M 2.94 mg/dL / W 3.44 mg/dL) - Q3 (M 3.16 mg/dL / W 3.62 mg/dL) - Q4 (M 3.38 mg/dL / W 3.84 mg/dL) - Q5 (M 3.75 mg/dL / W 4.24 mg/dL) |
|
Park (2022) |
No data |
Tertile 1 is considered as the reference. - T1 ( ≤ 3.2 mg/dL) - T2 (3.3-3.6 mg/dL) - T3 (≥ 3.7 mg/dL) |
|
Park (2020) |
No data |
Quartil 1 is considered as the reference. - Q1 (≤3.0 mg/dL) - Q2 (3.1 - 3.3 mg/dL) - Q3 (3.4-3.7 mg/dL) - Q4 ((≥ 3.8 mg/dl) |
|
Ling (2017) |
Hitachi 7600 biochemistry autoanalyzer (Roche Diagnostics, Basel, Switzerland) |
≤3.60 mg/dL vs >3.60 mg/dL |
|
Grønhøj (2016) |
Absorption photometry. |
OR is given per 0.31 mg/dL |
|
Park (2016) |
Automated clinical chemistry analyzer (Modular P analyzers; Roche Diagnostics, Tokyo, Japan) |
Quartil 1 is considered as the reference. - Q1 (≤ 3.2 mg/dL) - Q2 (>3.2 - ≤3.6 mg/dL) - Q3 (>3.6 - ≤4 mg/dL) - Q4 (> 4 mg/dL) |
|
Kwak (2014) |
Automated clinical chemistry analyzer (Modular P analyzers; Roche Diagnostics, Tokyo, Japan) |
Quartil 1 is considered as the reference. - Q1 (<3.4 mg/dL) - Q2 (3.2 – 3.6 mg/dL) - Q3 (3.6 – 3.8 mg/dL) - Q4 (> 3.9 mg/dL) |
|
Shin (2012) |
No data |
Quartil 1 is considered as the reference. - Q1 (similar 3.2 mg/dL) - Q2 (3.3-3.6 mg/dL) - Q3 (3.7-4 mg/dL) - Q4 (4.1 mg/dL) |
|
Cancela (2012) |
No data |
Tertile 1 is considered as the reference. Not available concentration of each tertil. |
|
Park (2011) |
No data |
Quartil 1 is considered as the reference. - Q1 (≤3.3 mg/dL) - Q2 (>3.3 - ≤3.6 mg/dL) - Q3 (>3.6 - ≤3.9 mg/dL) - Q4 (> 3.9 mg/dL) |
|
Foley (2009) |
SMAC 12 continuous-flow analyzer (Technicon Instruments Corp., Tarrytown, NY) |
Quartil 1 is considered as the reference. - Q1 (≤3.3 mg/dL) - Q2 (>3.3 - ≤3.6 mg/dL) - Q3 (>3.6 - ≤3.9 mg/dL) - Q4 (> 3.9 mg/dL) |
3. If possible consider what (%?) increase of Phosphate is associated to adverse outcomes or what range of values are associated to adverse outcomes. In your paper you report high levels of phosphate, anyway we have no idea of what range of levels have to be considered as a risk factor for the different adverse outcomes.
Thank you for your indication. As suggested, we have included this information in the tables above.

Reviewer 2 Report
Comments and Suggestions for Authors
This is an interesting Systematic review and Meta-analysis investigating the association of serum phosphate with cardiovascular mortality and subclinical coronary atherosclerosis. However, the following issues are needed to be addressed:
1) It should be acknowledged that the associations found in the current study cannot prove causality. The authors did not take into account the possibility that the associations of serum phosphate with cardiovascular mortality and subclinical coronary atherosclerosis that were found in the present study may reflect the impact of meat products and ultra-processed food, which are the main dietary source of phosphorus, on cardiovascular events. In this respect, serum phosphate could represent a biomarker, rather than a causal factor for cardiovascular events.
2) Similarly, weight loss is associated with changes in serum phosphate. Considering that changes in body weight can influence the incidence of cardiovasular events, the authors should report if the analyzed studies reported any body weight changes during follow up that may have confounded the overall results. The following study is relevant to this issue and can be added in the Reference list: "Not only the status of body weight and metabolic health matters for cardiovascular events, but also the temporal changes. Eur J Prev Cardiol. 2022;28(17):e25-e27.".
3) An important methodilogical limitation of the analyzed studies was that serum phosphate concentrations were determined at a single time point, without considering changes over time.
4) The size of the first paragraph of Results can be reduced to the final results of the literature search, since all the details are shown in Figure 1.
Comments on the Quality of English Language
This article is well-written. Minor proof reading is needed.
Author Response
Thank you for the detailed attention paid to our work.
Comments on the manuscript.
- It should be acknowledged that the associations found in the current study cannot prove causality. The authors did not take into account the possibility that the associations of serum phosphate with cardiovascular mortality and subclinical coronary atherosclerosis that were found in the present study may reflect the impact of meat products and ultra-processed food, which are the main dietary source of phosphorus, on cardiovascular events. In this respect, serum phosphate could represent a biomarker, rather than a causal factor for cardiovascular events.
Thank you for your comment. We apologize if this point is not clear enough. We agree that serum phosphate is not a causal factor of cardiovascular events. Indeed, it could be considered as a biomarker representing phosphorus intake from different dietary sources including, of course, meat products and ultra-processed foods. We are aware that the study designs (cross-sectional and cohort studies) do not allow us to assess causal effects, only associations. For this reason, we only describe associations in the text.
To make that clearer, we have changed 3 paragraphs:
- “However, we do not have enough evidence to draw conclusions about the variation in association by sex”. (Lines 412-413).
- “Serum phosphate concentrations can vary by up to 50% in a day, reflecting food intake [56]”. (Lines 431-432).
- “In addition, it is possible that dietary phosphorus derived from inorganic sources may have a greater influence on serum phosphate and parathyroid hormone [64]”. (Lines 448-450).
- Similarly, weight loss is associated with changes in serum phosphate. Considering that changes in body weight can influence the incidence of cardiovasular events, the authors should report if the analyzed studies reported any body weight changes during follow up that may have confounded the overall results. The following study is relevant to this issue and can be added in the Reference list: "Not only the status of body weight and metabolic health matters for cardiovascular events, but also the temporal changes. Eur J Prev Cardiol. 2022;28(17):e25-e27.".
Thank you for your indication. However, none of the studies has taken weight changes into account. We have added the following sentence with the reference at the end of the discussion:
“Also, in all cases, serum phosphate concentrations were measured at a single time point, without considering changes over time, which could lead to non-differential misclassification. Similarly, changes in weight loss [67] or changes in other confounding factors were not taken into account in the published articles, which could have led to regression dilution bias”. (Lines 529-533).
- An important methodological limitation of the analyzed studies was that serum phosphate concentrations were determined at a single time point, without considering changes over time.
Thank you for your comment. You are right. The fact that concentrations were measured at a single time point, is a limitation of the studies, that we have emphasized in discussion section:
“Also, in all cases, serum phosphate concentrations were determined at a single time point, without considering changes over time, which could lead to non-differential misclassification.”. (Lines 342-344).
- The size of the first paragraph of Results can be reduced to the final results of the literature search, since all the details are shown in Figure 1.
Done as requested. You can see the new paragraph:
“We identified 2,795 records through our search. After removing replicates, exclusions were made after reading the tittle or the abstract. A total of 56 reports were sought for retrieval and assessed for eligibility. Of these, 29 articles were excluded because did not meet inclusion criteria, and two articles because other outcomes were evaluated [33,34]. Finally, 25 quality studies were included in the review (all of them cohort or cross-sectional studies). No articles were excluded due to risk of bias (Figure 1)”. (Lines 186-191).

Round 2
Reviewer 1 Report
Comments and Suggestions for Authors
I recommend to consider also the sentence I have previously indicated. There is a reason for this recomendation, which could reinforce the value of your investigations.
If preventive actions(including the evaluation of phosphate alterations and metabolic correction) are initiated early, since childhood, there is a good chance to lower the premature CVD mortality and morbility. There are several conditions in childhood associated to phosphate increase, and the NIH has released recomendations to lower phospate intake.
So I suggest sto consider thie following sentence and the related reference.
"Effective algorithms to identify children at intermediate-high risk of future CV events have been proposed to promote the development of preventive programs which may personalize healthy lifestyles, behavioral modifications, and improved nutritional habits. [Reference: Ferraro S et al. Prediction of atherosclerotic cardiovascular risk in early childhood, Clinica Chimica Acta, Volume 552, 2024.] Your evidence could be considered for improving behavioural habits.
Author Response
Thank you for your attention, revision, and your suggestions.
Comments on the manuscript.
1. I recommend to consider also the sentence I have previously indicated. There is a reason for this recommendation, which could reinforce the value of your investigations.
If preventive actions (including the evaluation of phosphate alterations and metabolic correction) are initiated early, since childhood, there is a good chance to lower the premature CVD mortality and morbility. There are several conditions in childhood associated to phosphate increase, and the NIH has released recomendations to lower phospate intake.
So I suggest to consider the following sentence and the related reference.
"Effective algorithms to identify children at intermediate-high risk of future CV events have been proposed to promote the development of preventive programs which may personalize healthy lifestyles, behavioral modifications, and improved nutritional habits. [Reference: Ferraro S et al. Prediction of atherosclerotic cardiovascular risk in early childhood, Clinica Chimica Acta, Volume 552, 2024.] Your evidence could be considered for improving behavioural habits.
Thank you for your indication. As you suggested, we have included the following sentence in the discussion, that reinforces the value of our research for early life primary prevention:
“Therefore, effective algorithms have been proposed to identify children at intermediate-high risk of future CVD in order to promote the development of preventive programs which may personalize healthy lifestyles, behavioral modifications, and improved nutritional habits [59]”. (Lines 299-302).

Reviewer 2 Report
Comments and Suggestions for Authors
I appreciate the efforts of the authors to improve their manuscript. However, the Lines that are mentioned in their reply are not correct. Thus, I acannot find where the revisions are. Moreover, the authors should further revise the manuscript according to reviewers' comments.
Comments on the Quality of English LanguageThe manuscript nedds a thorough proof reading
Author Response
Thank you for your attention, revision, and your suggestions.
Comments on the manuscript.
1. I appreciate the efforts of the authors to improve their manuscript. However, the Lines that are mentioned in their reply are not correct. Thus, I cannot find where the revisions are. Moreover, the authors should further revise the manuscript according to reviewers' comments.
Thank you for your comment. We apologize if you were unable to locate the lines correctly, there has been an error. I am quoting the sentences next to the corresponding lines below.
- “We identified 2,795 records through our search. After removing replicates, exclusions were made after reading the tittle or the abstract. A total of 56 reports were sought for retrieval and assessed for eligibility. Of these, 29 articles were excluded because did not meet inclusion criteria, and two articles because other outcomes were evaluated [32,33]. Finally, 25 quality studies were included in the review (all of them cohort or cross-sectional studies). No articles were excluded due to risk of bias (Figure 1)”. (Lines 147-152).
- “However, we do not have enough evidence to draw conclusions about the variation in association by sex”. (Lines 274-275).
- “Serum phosphate concentrations can vary by up to 50% in a day, reflecting food intake [54]”. (Lines 289-290).
- “In addition, it is possible that dietary phosphorus derived from inorganic sources may have a greater influence on serum phosphate and parathyroid hormone [62]”. (Lines 310-312).
- “Also, in all cases, serum phosphate concentrations were measured at a single time point, without considering changes over time, which could lead to non-differential misclassification. Similarly, changes in weight loss [66] or changes in other confounding factors were not taken into account in the published articles, which could have led to regression dilution bias”. (Lines 388-392).
In addition, according to your rating we have thought it convenient to modify three paragraphs of the introduction and the discussion, improving the quality of this review:
- “Previously, only two meta-analyses assessing the association between serum phosphate and cardiovascular risk have been published. One of them examined the association between elevated fasting serum phosphate concentration and the risk of death and CVD in individuals with chronic kidney disease (CKD) [23]. Therefore, the results could not be generalized to the general population. The second one assessed the association of serum phosphate with all-cause and cardiovascular mortality in the general population; however, it is based on a limited number of studies [24]. The study of this association is of great interest, recently published articles suggest that even an elevated serum phosphate concentration within the normal range might contribute to increased cardiovascular risk independently of kidney function [25–26]”. (Lines 70-79).
- “In contrast, inorganic phosphorus is more bioavailable, being almost completely absorbed in the digestive tract as they do not require enzymatic digestion and dissociate rapidly in the stomach acids; being composed of inorganic phosphate the phosphorus additives [15,17–20]”. (Lines 307-310).
- “Regarding CVD and serum phosphate, both postprandial elevation of serum phosphate and continued elevation of serum phosphate should be considered in the development of CVD [64]. Certain in vitro studies have shown that hyperphosphatemia induces phenotypic conversion of vascular smooth muscle cells into osteoblast-like cells expressing biochemical markers characteristic of bone tissue, resulting in vascular calcification in humans [65]. In addition, both in vitro and in vivo studies have demonstrated that high dietary phosphorus intake causes endothelial dysfunction within a short period, suggesting that the elevation of serum phosphate concentration due to dietary phosphorus load may be a risk factor for endothelial dysfunction in both healthy individuals and those with CKD [64]. However, the causal relationship between higher serum phosphate concentrations and coronary atherosclerosis has not been fully explained yet”. (Lines 318-328)
Comments on the Quality of English Language
1. The manuscript nedds a thorough proof reading.
Thank you for your suggestion. This article has been reviewed by a native English speaker.
